# Current Understanding of the Pathophysiology and Novel Treatments of Interstitial Cystitis/Bladder Pain Syndrome

**DOI:** 10.3390/biomedicines10102380

**Published:** 2022-09-23

**Authors:** Jia-Fong Jhang, Yuan-Hong Jiang, Hann-Chorng Kuo

**Affiliations:** Department of Urology, Hualien Tzu Chi Hospital, Buddhist Tzu Chi Medical Foundation, Buddhist Tzu Chi University, Hualien 970, Taiwan

**Keywords:** bladder pain, cystitis, pathophysiology, intravesical therapy

## Abstract

The pathophysiology of interstitial cystitis/bladder pain syndrome (IC/BPS) is multifactorial. Identifying the clinical characteristics and cystoscopic findings of bladder-centered IC/BPS facilitates optimal treatment strategies targeting the diseased urinary bladder. Patients with Hunner’s lesion (HIC) and without Hunner’s lesion (NHIC) should be treated differently. Based on the histopathological findings, NHIC can be treated with intravesical instillation of urothelial protective agents, such as hyaluronic acid, to cover the urothelial defects. In non-responders, chronic inflammation and higher urothelial dysfunction can be treated with intravesical botulinum toxin A injection, platelet-rich plasma injection, or low-energy shock wave treatment to reduce inflammation, increase tissue regeneration, and improve the urothelial barrier. Patients with HIC should be treated with electrocauterization first; augmentation enterocystoplasty should only be used in end-stage HIC when the contracted bladder is refractory to other treatments. The antiviral agent, valacyclovir, can be used in patients with HIC, small bladder capacity, and high-grade glomerulations. In addition, behavioral modification is always recommended from the beginning of treatment. Treatment with cognitive behavioral therapy interventions in combination with bladder therapy can reduce anxiety and improve treatment outcomes. Herein, recent advances in the pathophysiology and novel treatments for IC/BPS are reviewed.

## 1. Introduction

Interstitial cystitis/bladder pain syndrome (IC/BPS) is characterized by bladder pain and urinary frequency, with or without urgency, and nocturia. Because the underlying pathophysiology is unclear, definitive treatment has not been achieved. The known pathological findings from the bladders of IC/BPS patients included urothelial denudation, chronic inflammation, and sensory afferent activation [1]. Some patients also have multiple somatic complaints and functional somatic syndrome [2]. Currently, there is no durable treatment for this disease; however, novel treatments have been designed based on new discoveries in the pathophysiology of IC/BPS. This article reviewed the recent understanding of the pathophysiology and treatments of IC/BPS.

The first report of IC was based on severe bladder pain and the cystoscopic finding of “ulceration” in the bladder wall. Fifty years later, diffuse glomerulations were noted after cystoscopic hydrodistention in patients with IC, and “early IC” or “non-ulcer IC” was proposed [3]. However, the characteristic glomerulations are not routinely observed during cystoscopic hydrodistention in patients with IC/BPS symptoms. Therefore, “bladder pain syndrome” was suggested for patients with symptoms of bladder pain and increased urinary frequency but no bladder ulcers [4]. “Hypersensitive bladder syndrome” was proposed in the recent Asian IC guidelines [5]. The heterogeneous clinical presentations of IC/BPS cause ambiguous classifications and difficulties in treating and curing this bladder disorder.

## 2. Definition and Diagnosis of IC/BPS

The common clinical symptoms of IC/BPS include bladder pain, urinary frequency, the presence of Hunner’s lesions, and diffuse glomerulations under cystoscopic hydrodistention. The most common pathological findings are urothelial denudation and bladder inflammation [5,6]. The United States National Institute of Diabetes and Digestive and Kidney Diseases suggests that IC be diagnosed by exclusion. However, this definition is considered too strict for the clinical enrollment and early treatment of IC/BPS patients [7]. The American Urological Association guidelines suggest that IC be considered and treated in patients with characteristic symptoms for more than six weeks that are refractory to treatment [8]. The European Society for the Study of Interstitial Cystitis (ESSIC) classified IC/BPS subtypes and recommends treatment modalities including lifestyle modification, medication for bladder pain, bladder glycosaminoglycan replacement, intravesical botulinum toxin A (Botox) injection, neuromodulation, and surgical intervention [9]. However, the long-term treatment outcomes for different subtypes of IC/BPS patients are inconsistent. The underlying pathophysiology might affect the treatment outcomes.

## 3. Possible Pathogenesis and Principles of Treatment of IC/BPS

The prominent pathological findings of IC/BPS bladders include chronic inflammation and urothelial denudation. Inflammation is higher in bladders with Hunner’s lesions (HIC) compared with inflammation in non-Hunner’s IC (NHIC) [10,11]. The severity of bladder wall inflammation is highly associated with the severity of IC symptoms, glomerulation grade, and maximal bladder capacity under cystoscopic hydrodistention in bladder-centered IC/BPS patients, which indicates the pathological lesions are located inside the urinary bladder [10,12]. Increased microvascular endothelial cell apoptosis in IC bladders causes glomerulations [13], and impaired urothelial homeostasis is associated with chronic bladder inflammation [1]. Chronic pain in patients with IC/BPS is likely caused by central nervous system (CNS) sensitization and activation of sensory afferent nerves [14]. Bladder epithelial dysfunction, mast cell activation, neurogenic inflammation, autoimmunity, and occult infection may also cause bladder pain.

The treatment of IC/BPS should follow the possible pathophysiology and specific clinical presentation. Because the actual etiology and diagnosis of IC/BPS remain unclear, treatments usually target the pathogenesis, including the post-infection autoimmune process, mast cell activation, urothelial dysfunction, and neurogenic inflammation. Bladder dysfunction might originate from chronic inflammation, causing urothelial defects and more tissue reactions, subsequently leading to inflammation of the CNS [15].

In recent decades, the following treatments for IC/BPS have been recommended: (1) replenishment of the glycosaminoglycan layer to control urothelial defects, (2) administration of amitriptyline or imipramine to inhibit neurological hyperactivity, (3) suppression of allergies by antihistamines, and (4) non-steroid anti-inflammatory drugs for pain control. Intravesical treatments with heparin, hyaluronic acid, chondroitin sulfate, bacillus Calmette-Guerin, dimethylsulfoxide, and resiniferatoxin have been effective in treating IC/BPS, but the effects were not durable. Multidisciplinary therapy with medical treatment, intravesical treatment, and psychiatric consultation may achieve synergistic effects with better outcomes [16]. For IC/BPS patient’s refractory to the initial treatments, intravesical Botox injections or sacral neuromodulation may provide long-term symptomatic relief. Surgical treatment, such as bladder augmentation, is only recommended for IC/BPS patients who do not respond to medical therapy [4].

## 4. Treatment Targeting Urothelial Dysfunction and Regenerative Deficit in IC/BPS

### 4.1. Inflammation Is the Main Pathophysiology of IC/BPS

Aberrant differentiation of bladder urothelium in IC/BPS bladders leads to decreased production of cell surface protectants such as proteoglycans, adhesion proteins E-cadherin, tight junction proteins zonula occludens-1, and bacterial defense molecules [17]. A previous study found apoptotic signaling molecules were increased in the bladder of IC/BPS, including Bad, Bax, and cleaved caspase-3 [18]. Intravesical Botox injections may reduce the levels of these apoptotic proteins. The upregulation of inflammatory signals may induce apoptosis in IC/BPS bladders, possibly through TNF-α and p38 MAPK signaling pathways [18]. The expressions of inflammatory cells, focal lymphoid aggregates, and B-cell and T-cell markers are increased in IC/BPS bladders [19]. The serum levels of pro-inflammatory proteins are also increased in patients with IC/BPS, suggesting the main pathophysiology of IC/BPS is inflammation of the urinary bladder [20].

In addition to the bladder pathophysiology found in IC/BPS patients, which may account for the clinical presentations of bladder-centered IC/BPS, many somatic and functional disorders coexist with IC/BPS. Systemic inflammatory diseases may also play important roles in the pathogenesis of IC/BPS [20]. IC/BPS patients with low maximal bladder capacity and high histological inflammation levels in the bladder have a distinct bladder-centric phenotype [21]. The IC/BPS symptoms might be caused or aggravated by inflammatory bladder conditions, mental factors, such as stress disorders, and medical comorbidities [22]. Not all IC/BPS patients have the same symptoms and bladder histopathological findings. Comorbid systemic diseases and mental illnesses are frequently noted in patients with IC/BPS [2,23]. Functional somatic syndrome is closely associated with IC/BPS [24]. Therefore, IC/BPS may involve heterogeneous patient subgroups that present with different clinical symptoms, urodynamic results, and cystoscopic findings [25].

### 4.2. Deficits in Bladder Epithelial Cell Differentiation and Maturation in IC/BPS

Histologically, partial or complete deficits in urothelial cells are prominent in IC/BPS bladders [9,16]. Expression levels of the tight junction protein, zonula occludens-1, and the adhesive protein, E-Cadherin, are also reduced in IC/BPS bladders [25,26]. These anatomical and functional defects of the bladder urothelium are important pathogenic mechanisms of IC/BPS [27]. IC/BPS bladders exhibit differential expression deficits in bladder epithelial cell differentiation and maturation markers, including cytokeratin (CK)-5, CK-14, and CK-20 [28,29,30]. CK dysregulation in the bladder epithelium may be associated with urothelial dysfunction and the clinical characteristics of IC/BPS (Figure 1). The mature cell marker, CK20, and the cell proliferation proteins, P63 and FGFR4, are lower in patients with HIC or NHIC and grade 3 glomerulations compared with other subtypes of NHIC and controls; the immature cell marker, CK5, and the apoptotic protein, BAX, are higher in the same subtypes [31]. Urothelial deficits in IC/BPS bladders may cause barrier dysfunction, resulting in bladder pain and hypersensitivity while the bladder is full. The abnormal CK levels and epithelial cell proliferation proteins are likely caused by persistent chronic inflammation in the bladders of patients with IC/BPS [19].

### 4.3. Increased Urothelium Cell Apoptosis Is Mediated by Inflammation in IC/BPS

Histological studies of partial cystectomy specimens from HIC bladders show high-grade inflammation and germinal center formation [32,33]. In one study, all HIC bladders had chronic inflammation [34]. Our findings demonstrated that the grade of bladder glomerulations is associated with suburothelial granulation but not inflammation or urothelium deficits. These findings agree with previous studies showing that bladder glomerulations are associated with higher vascular endothelial growth factor (VEGF) expression, suggesting that glomerulations are due to neovascularization in IC/BPS bladders [35,36]. In addition, eosinophil and plasma cell infiltrations are also found in IC/BPS bladders [37,38]. The nerve hyperplasia marker, S-100, was noted in all IC/BPS bladders [39]. We recently showed that nerve hyperplasia was present in 100% of HIC and 95% of NHIC bladders, as indicated by staining of neuron-specific protein gene product 9.5 [40]. Thus, nerve hyperplasia is very common in IC/BPS bladders, which may explain why patients with IC/BPS have increased urinary frequency and bladder pain without a full bladder.

### 4.4. Intravesical Injections of Platelet-Rich Plasma to Improve Urothelial Regeneration for IC/BPS

Chronic inflammation in the bladders of patients with IC/BPS causes urothelial deficits through increased urothelial cell apoptosis, nociceptive receptors upregulation, and mast cell activation, resulting in bladder pain [6,18]. Persistent bladder inflammation damages urothelial integrity and increases urothelial barrier permeability [1,41]. Therefore, treatments for IC/BPS should target not only chronic inflammation but also the regenerative ability of the urothelium and the barrier function of the bladder epithelium to improve bladder symptoms.

Platelet-rich plasma (PRP) is widely used as a regenerative therapy [42]. PRP is rich in growth factors and cytokines that can modulate the inflammatory process and facilitate wound healing to reduce neuropathic pain [43,44]. IC/BPS symptoms correspond to the severity of bladder wall inflammation, which causes poor basal cell proliferation, defective umbrella cell integrity, and decreased barrier function [1,18,45]. The defective urothelium in IC/BPS is similar to an unhealed wound in the bladder mucosa. Therefore, the use of PRP to promote mucosal wound healing and eradicate residual inflammation in the bladder is rational [44]. Urothelial cell proliferation, cytoskeleton markers, and barrier function protein expression increase after PRP treatment in IC/BPS bladders [46].

We conducted a study to determine the therapeutic efficacy of intravesical PRP injections in IC/BPS patients [47]. Forty patients received four monthly PRP injections and follow-up visits. Global response assessment scores improved after the first PRP injection, and the therapeutic effect persisted up to three months after the fourth PRP injection; the success rate was 67.5%. IC symptom scores and bladder pain scores decreased. Bladder capacity increased, but PVR did not change after PRP treatment, and no UTIs or dysuria occurred. Thus, repeated autologous PRP injections may be effective in patients with IC/BPS refractory to conventional therapy.

After four PRP injections in patients with IC/BPS, urinary levels of NGF, MMP-13, and VEGF decreased, and PDGF-AB levels significantly increased in responders, suggesting that PRP injections decrease inflammation, resulting in improvements in bladder pain and frequency episodes [48]. In addition, the expression of the urothelial barrier function protein and the urothelial cell proliferation protein increased after repeated PRP injections [49]. Thus, intravesical PRP injections may improve urothelial regeneration and differentiation to recreate a healthy urothelial barrier in IC/BPS bladders [50]. Repeated PRP intravesical injections improve IC/BPS symptoms by promoting the recovery of urothelial ultrastructural defects [51].

## 5. Treatment Targeting Urothelial Dysfunction and Barrier Deficit in IC/BPS

The bladder urothelium acts as a barrier, preventing urine, solutes, and toxins from crossing the cell surface. Surface mucus secreted onto the bladder epithelium, including glycosaminoglycans (GAG), is critical in maintaining normal urothelial function [52,53,54]. GAG, a product of normal mature bladder urothelium, protects the bladder epithelium from bacterial adhesion and infection and penetration by toxic substances from the urine [55]. 

### 5.1. Bladder Urothelial and Barrier Deficits in IC/BPS

Deficits in the barrier function of the bladder urothelium result in the migration of urinary metabolites and solutes, including hydrogen or potassium. These solutes and metabolites may depolarize sensory nerves, provoke bladder contractile reflexes, or cause tissue injury [56,57]. Patients with IC/BPS often have a leaky epithelium, and potassium ions can diffuse through the defective cell barrier and provoke nerve terminals, causing urinary urgency, frequency, and bladder pain. Instilling high concentration potassium chloride solution into the bladder elicits pain and urgency, demonstrating the presence of urothelial leakage in patients with IC/BPS [54]. Intravesical sulfated polysaccharide pentosan polysulfate can restore the defective urothelium to its normal condition [1].

Electron microscopic (EM) studies also confirmed various ultrastructure changes in the urothelium of IC/BPS bladders, including complex junctional deficits, pleomorphisms of urothelial cells, loss of umbrella cells, loss of surface microvilli, and increases in mast cells (Figure 2) [58,59,60]. Loss of umbrella cells and reduced cell membrane microplicae are significantly associated with the clinical symptoms of IC/BPS [41]. We showed that urothelial cell numbers are significantly reduced, and the umbrella cell barrier integrity is lost in HIC bladders. Suburothelial inflammatory cell infiltration is also commonly found in IC/BPS bladders, especially in patients with high-grade glomerulation after hydrodistention [41]. These urothelium defects, shown by EM, are strongly associated with clinical symptom severity in patients with IC/BPS, suggesting that urothelial dysfunction is a crucial part of the pathophysiology of IC/BPS.

### 5.2. Intravesical Instillation of Hyaluronic Acid to Replenish Urothelail Defects

Based on urothelial dysfunction studies, aberrant differentiation and defective synthesis of surface proteoglycans and barrier proteins contribute to IC/BPS. Therefore, intravesical therapy with glycosaminoglycan replacement is widely applied for IC/BPS treatment [61].

Hyaluronic acid (HA) is a non-sulfated mucopolysaccharide component of the GAG layer. It is present on the bladder wall surface to prevent urine and solutes from crossing the cell membrane. Intravesical instillation of HA weekly for four weeks, followed by monthly treatments, provoked a 71% positive response rate at week 12 that was maintained up to week 20 [62]. Improvements in pain and frequency were observed in a small cohort of IC/BPS patients treated with intravesical HA instillations, and 55% of patients preferred to continue this treatment for symptom relief [63]. The treatment efficacy of HA instillations was demonstrated during cystoscopic hydrodistention in patients with severe IC/BPS symptoms [64]. Long-term symptom remission at the 5-year follow-up was observed in 50% of IC/BPS patients with intravesical HA instillations. Patients who had symptom recurrence improved with HA maintenance therapy [65]. Our previous randomized trial showed that four weekly treatments with HA instillations followed by five monthly treatments or nine biweekly treatments significantly improved symptoms and quality of life in patients with IC/BPS [66].

## 6. Treatment Targeting Neurogenic Inflammation of IC/BPS

Neurogenic inflammation is involved in the pathophysiology of IC/BPS [67]. The levels of immunoreactive substance P and nerve growth factor (NGF) increase in bladder tissue and the urine of IC/BPS patients compared with healthy control [68,69]. NGF levels are significantly higher in the bladders of patients with IC/BPS compared to healthy controls [70]. Furthermore, intravesical NGF instillation causes bladder inflammation, resulting in acute afferent nerve hyperactivity and long-term plasticity, and may lead to lower nociceptive and mechanoreceptive afferent fiber thresholds [71,72,73]. Increased NGF levels in bladder tissue probably initiate signals along the afferent fibers, from the bladder to the dorsal root ganglion of the spinal cord, that trigger bladder hypersensitivity to chemical or mechanical stimulation [74,75]. Intravesical injection of Botox may reduce bladder pain in IC/BPS patients by lowering NGF levels in the bladder [76]. In the cyclophosphamide-induced cystitis rat model, muscarinic M5 receptors increased in the urothelium and endothelial nitric oxide synthase was up-regulated, and the inflamed bladder strips were less inhibited by 4-DAMP inhibition, suggesting the urothelium plays a significant role in the pathogenesis of IC/BPS [77]. A recent study also revealed that HMGB1 promotes somatic and visceral pain signals [78]. An animal model of cystitis demonstrated that bladder pain involved ATP-dependent HMGB1 release from macrophages, causing nociceptor excitation and resulting in bladder pain [79]. This pathophysiology might provide novel therapy targeting bladder pain in IC/BPS.

The C-fibers and purinergic receptors P2X3 are involved in neurogenic inflammation in several functional diseases and may also be involved in neuroinflammation in IC/BPS. These fibers are important in pain response and the control of bladder volume reflexes [80]. The P2X3 receptors are localized to suburothelial C-fibers along with many sensory receptors responsible for controlling the bladder sensory threshold functional bladder capacity [81]. Damage to the bladder urothelium by trauma, toxins, or chemical irritation induces a chain reaction of inflammation, resulting in pain and urinary frequency in IC/BPS bladders [82]. Thus, IC/BPS may be induced sequentially by urothelium injury, suburothelial inflammation, inflammatory cell infiltration, chronic scar formation in the suburothelium, and increased inflammation in the CNS [83]. An injury or inflammatory event in the urinary bladder activates an acute inflammatory reaction, resulting in an inflammatory imprint in the spinal cord and cerebral cortex. During the initial bladder insult, the acute inflammatory reaction causes symptoms characteristic of IC/BPS. If the insult does not continue, the immune system can solve the inflammation and relieve the bladder symptoms. Following persistent bladder insults and inflammation, suburothelial inflammation may impair urothelial cell proliferation and maturation, leading to barrier deficits and, eventually, long-term bladder inflammation. Chronic pelvic pain syndrome may result from the presence of a high level of inflammation in the CNS and referred pain [84].

### 6.1. Effects of Intravesical Botox Treatment on Refractory IC/BPS

Chronic inflammation of the urinary bladder may be the underlying pathophysiology of IC/BPS [85]. Chronic unsolved inflammation in the bladder suburothelium results in impaired proliferation and maturation of urothelial cells, urothelial denudation, deficits of tight junction and barrier proteins, and increased bladder wall permeability [1,86,87,88,89]. Bladder pain can be reduced by detrusor Botox injections in IC/BPS patients [90]. Repeated Botox injections can increase the functional capacity of the bladder and improve IC bladder symptoms in responders [91]. Botox treatment reduces bladder inflammation and improves cell proliferation, leading to symptomatic relief in responders [23,91]. The early clinical trial demonstrated that intravesical Botox injections alleviate bladder pain and reduce urinary frequency and nocturia episodes in patients with IC/BPS [92]. The glomerulation grade also decreases, and the maximal bladder capacity increases after cystoscopic hydrodistention in response to Botox injections (Figure 3).

A randomized, double-blind, placebo-controlled clinical trial demonstrated that 100 U of Botox is safe and effective for treating IC/BPS and results in a significant reduction in bladder pain and an improvement in bladder capacity compared with bladder pain and capacity in the placebo group [93]. Botox had anti-inflammatory effects in an animal cystitis model [94]. Botox injections decrease the expression of NGF and relieve bladder pain in patients with IC/BPS [15]. The therapeutic effects of Botox involve both sensory and motor nerves. Intravesical Botox injections may reduce sensory urgency, detrusor contractility, and the release of noxious neurotransmitters in patients with IC/BPS [82]. Therefore, Botox can reduce urinary urgency and increase functional bladder capacity [95].

Based on the evidence presented above, Botox intravesical injection is recommended as the fourth-line treatment in the American Urological Association clinical guidelines for IC/BPS [96]. However, a warning about urinary retention after Botox injections is included in the guidelines. We found that the post-void residual (PVR) volume significantly increased after Botox injections in both responder and non-responder IC/BPS patients. However, only 10% of patients complained of dysuria after Botox injections, and no patients experienced acute urinary retention, indicating that Botox is a safe treatment for IC/BPS [90].

The reported success rate of intravesical Botox injection for IC/BPS patients was approximately 60% [93]. However, the factors that predict successful outcomes and the risk factors for difficult urination after Botox injections in IC/BPS patients have not been determined. The long-term efficacy of Botox treatment and the suitable doses and injection techniques have not been well established [97]. Different IC/BPS phenotypes may have different Botox treatment outcomes. In a recent study, a high baseline maximal bladder capacity, but not glomerulation grade or urodynamic parameters, predicted satisfactory outcomes after Botox injections in IC/BPS patients [98]. Nevertheless, repeated intravesical Botox injections seem to be necessary for long-term symptom improvement [90].

### 6.2. Liposome-Encapsulated Botox Intravesical Instillation in Treating IC/BPS

Although intravesical Botox injections are effective in the treatment of IC/BPS, the necessity of anesthesia, potential complications of intravesical injections, and dysuria or urinary tract infections (UTIs) after treatment still limit the widespread application of this treatment. Liposomes are lipid vesicles composed of phospholipid bilayers. Drugs or molecules can be incorporated into liposomes to facilitate passage across the cell membrane through endocytosis [99,100]. Intravesical instillation of empty liposomes has therapeutic effects in patients with IC/BPS. Liposomes may coat the bladder urothelium and enhance the barrier function of the defective urothelium in IC/BPS bladders [101]. A previous study using liposome-encapsulated Botox (Lipotoxin) demonstrated that the synaptosome-associated protein (SNAP) -25 can be cleaved in the bladder urothelium [102].

Intravesical instillation of liposome-encapsulated Botox is an effective treatment for overactive bladder [103]. Urinary frequency and urgency significantly improved in patients treated with Lipotoxin compared with frequency and urgency in the placebo group. Furthermore, Lipotoxin caused no adverse events, such as UTIs, dysuria, or urinary retention. In a pilot clinical trial using Lipotoxin instillation to treat IC/BPS, we also found a statistically significant decrease in the O’Leary-Sant symptom scores and bladder pain severity and an increase in the global response assessment in patients treated with Lipotoxin. However, Lipotoxin was not superior to the therapeutic efficacy of Botox or the placebo groups. In the analysis, intravesical Lipotoxin instillation was associated with a decrease in symptoms from baseline in patients with moderate to severe IC/BPS [104].

### 6.3. Low-Energy Shock Wave Bladder Treatment for IC/BPS

Current evidence indicates that IC/BPS might result from neurogenic inflammation causing bladder afferent nerve hyperactivity and resulting in bladder hypersensitivity [104]. Chronic bladder inflammation also impairs normal urothelial proliferation and maturation and limits the regeneration of urothelial defects in IC/BPS [17,105].

Low-energy extracorporeal shock wave (LESW) therapy has been used to treat inflammation and improve soft tissue regeneration in tendonitis, ischemic cardiovascular disorders, and male erectile dysfunction [106,107,108]. LESW therapy generates growth factors and cytokines and activates progenitor cells, culminating in endothelial regeneration in the injured tissue [106]. LESW can also modulate tissue inflammation [109]. In a rat model of cyclophosphamide cystitis, LESW treatment inhibited the expression of NGF, IL-6, and COX-2, and reduced bladder pain, hyperactivity, and tissue inflammation [110]. LESW also reduced symptoms in human chronic prostatitis and pelvic pain syndrome [111,112]. Thus, LESW may also have therapeutic effects in patients with IC/BPS.

LESW elicits anti-inflammatory effects and pain relief by increasing local blood circulation at the treatment site [110,113]. In addition, LESW inhibits apoptosis and enhances cell proliferation, which may facilitate urothelial regeneration and epithelium repair in the bladders of patients with IC/BPS [114]. In a recent phase II, multicenter, double-blind, randomized, placebo-controlled clinical trial, no significant differences in IC symptoms or bladder pain were detected between the LESW-treated and placebo groups during follow-up [115]. Interestingly, pain scores (visual analog scale, VAS ≥ 3) were reduced in more than 50% of patients with LESW, and the reduction rate was significantly higher in the LESW-treated group compared with the placebo group. Frequency episodes also improved in this study.

## 7. Treatment of IC/BPS with Hunner’s Lesion

HIC, an IC/BPS subtype distinct from NHIC, is characterized by severe inflammation, lymphoid follicles, and fibrosis. In radiology studies of HIC bladders, focal and diffuse bladder wall thickening was associated with smaller bladder capacity, higher glomerulation grade, and the presence of Hunner’s lesions [116]. The clinical guidelines for IC/BPS suggest that HIC should be more aggressively treated than NHIC [4]. The recommended treatment for HIC includes transurethral electrocauterization or laser fulguration of the Hunner’s lesions, which may provide long-lasting therapeutic effects [117,118,119]. In cases of HIC recurrence with a contracted bladder and severe bladder pain, surgical interventions, such as partial cystectomy with or without bladder augmentation or complete cystectomy with an ileal conduit might be necessary as a last resort for HIC refractory to all conservative or less invasive treatments [120]. Patients usually achieve pain relief after surgery, but patients should be informed about the surgical morbidity and postoperative complications before the procedure. Occasionally, pelvic pain, which may stem from CNS sensitization, persists after partial cystectomy with augmentation in chronic HIC patients [121]. Long-term postoperative monitoring for recurrent UTIs and difficult urination is necessary after partial cystectomy and enterocystoplasty [122].

## 8. Antiviral Treatment for IC/BPS

The predominant pathological findings from bladder biopsies in patients with IC/BPS include urothelial denudation and increased inflammatory cell infiltration [10,11]. Bladder wall inflammation is more pronounced in HIC patients compared to that seen in NHIC patients. The histopathology of HIC bladder includes dense inflammatory changes and the development of lymphoid follicles, indicating that microorganism infection might contribute to HIC. In the AUA guidelines for IC/BPS, patients with bacterial bladder infections should be excluded from the diagnosis of IC/BPS; however, viral infections are not listed in the exclusion criteria [4]. We were the first to report the presence of Epstein–Barr virus (EBV) infections in the bladders of patients clinically diagnosed with IC/BPS, detected using in situ hybridization of EBV-encoded small RNAs and quantitative polymerase chain reaction for the EBV BNRF1 gene [33]. EBV infections were detected in 87.5% of HIC patients versus 17.4% of NHIC patients. This important finding indicates that EBV is a candidate pathogen responsible for chronic inflammation and the cicatrized bladder condition in HIC patients [33].

EBV, which can shuttle between different cell types and switch between latent and lytic infection cycles, has been implicated in several autoimmune diseases, including systemic lupus erythematosus, Sjögren’s syndrome, and rheumatoid arthritis [123]. Chronic or recurrent EBV infection of B cells leads to a persistent immune response [124]. Anti-herpes virus drugs, such as acyclovir and valacyclovir, inhibit herpes virus replication in the lytic cycle [125] to suppress EBV replication [126,127]. Several systemic diseases associated with EBV infections are effectively treated with valacyclovir [126,128,129]. Thus, chronic inflammation in HIC bladders may also be responsive to valacyclovir if EBV lytic infection is present in the bladder wall.

In a recent unpublished clinical trial, 10 of 13 EBER-positive bladder specimens (76.9%) were positive for EBV nuclear antigen 1 (EBNA1, latency infection marker), six specimens (46.2%) were positive for latent membrane protein 1 (LMP1, latency infection marker), and four specimens (19.0%) were positive for immediate-early protein BRLF1 (lytic infection marker). Moreover, all BRLF1-positive bladders were positive for EBNA1, indicating the concurrence of both latent and lytic EBV infections. Both B and T cells can host EBV infections. Thus, oral valacyclovir (500 mg twice a day for one month) was prescribed for four HIC patients and 24 NHIC patients. This treatment significantly improved VAS scores after the first week of treatment and continuously improved VAS scores after the first month. Ten patients (35.7%) responded to one month of treatment with a VAS reduction of ≥ 3. The four patients with HIC were all responders, and two were pain-free after treatment. EBV was undetectable in urine samples after valacyclovir treatment. [Jhang JF et al. unpublished data] Based on this pilot study, antiviral drug treatment is promising in patients with HIC refractory to conventional therapy.

## 9. Pelvic Floor Manual Therapy and Psychiatric Consultation to Treat IC/BPS

Patients with IC/BPS are at high risk of comorbid autoimmune or neurological diseases [2,130,131]. A distinct phenotype of IC/BPS has multiple psychosomatic sensitivities and disorders [24]. Although most IC/BPS patients are bladder-centered, crosstalk between pelvic organs and the bladder may be the pathophysiology responsible for the non-bladder-centered IC/BPS subtype. This subtype of IC/BPS is characterized by pelvic organ pain, irritable bowel syndrome, and myofascial pain, in addition to bladder symptoms [132]. Small bladder capacity is a marker for bladder-centered IC, whereas patients with higher bladder capacity may exhibit psychosomatic syndrome, more depression, and irritable bowel syndrome [133]. We demonstrated that IC/BPS patients with glomerulation grade 0 or 1 and maximal bladder capacity ≥ 760 ml had more comorbidities, suggesting the coexistence of comorbidities outside the bladder.

Currently, no effective and long-lasting treatments for IC/BPS exist. Patients with IC/BPS have lower urinary tract dysfunction that causes psychological stress [12,134]. Nevertheless, knowledge of the pathophysiology of IC/BPS has evolved, and novel treatments are being developed and tested. Classification of IC/BPS patients by their clinical characteristics, urodynamic results, and urinary biomarkers may provide therapeutic directions for specific IC/BPS subgroups [33,135]. The standard treatment algorithm for IC/BPS should begin with lifestyle modification, followed by therapy targeting bladder pathology. In IC/BPS patients with high anxiety, a combination of bladder therapy and psychological adjustment might improve symptoms [136]. In addition, psychological stress is usually associated with IC symptom exacerbation, including bladder pain and urinary frequency and urgency [137]. Psychiatric consultation should include bladder education, behavioral modification, coping mechanisms for stress, and increased tolerance to bladder conditions for patients with IC/BPS. The psychiatric consultation should be incorporated into a multimodal treatment plan for patients with IC/BPS. Thus, effective treatment for IC/BPS patients with moderate to severe anxiety should include both bladder therapy and psychiatric consultation to provide IC symptom relief and quality of life improvements [8].

Figure 4 demonstrates the current understanding of the pathophysiology of IC/BPS, clinical characteristics, and conventional treatments, as well as novel therapies targeting each possible pathophysiology of IC/BPS. 

## 10. Conclusions

The fundamental pathophysiology of IC/BPS includes chronic bladder inflammation, increased urothelial cell apoptosis, decreased urothelial proliferation and differentiation, and defects in the urothelial barrier. The pathogenesis of HIC is similar to NHIC, but the disease severity and treatment strategy differ. In NHIC, chronic inflammation and high urothelial dysfunction are treated with intravesical Botox injection, PRP injection, or LESW treatment to eliminate inflammation, facilitate tissue regeneration, and improve urothelial barrier function. Patients with HIC should be treated with electrocauterization first, and invasive surgery should be reserved for end-stage bladder disease. The antiviral agent, valacyclovir, may be tried in patients with HIC, small bladder capacity, and high-grade glomerulations. In addition, cognitive behavioral therapeutic intervention should be tried in combination with bladder therapy to reduce anxiety and improve treatment outcomes.

## Figures and Tables

**Figure 1 biomedicines-10-02380-f001:**
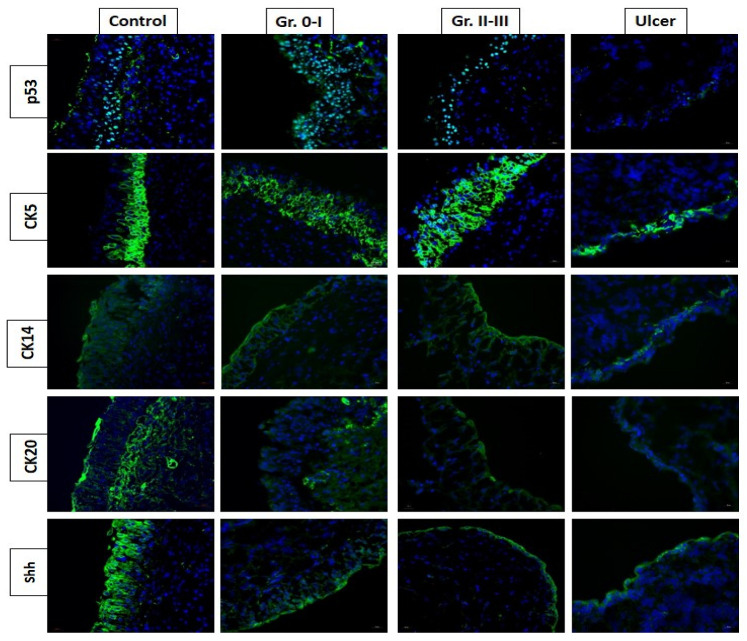
The expression of bladder urothelial proliferation and differentiation markers in different interstitial cystitis (IC)/bladder pain syndrome subtypes. The expression of sonic hedgehog (Shh) and cytokeratin (CK) 20 proteins are decreased in ulcer type IC and non-ulcer IC with high-grade glomerulations (Gr).

**Figure 2 biomedicines-10-02380-f002:**
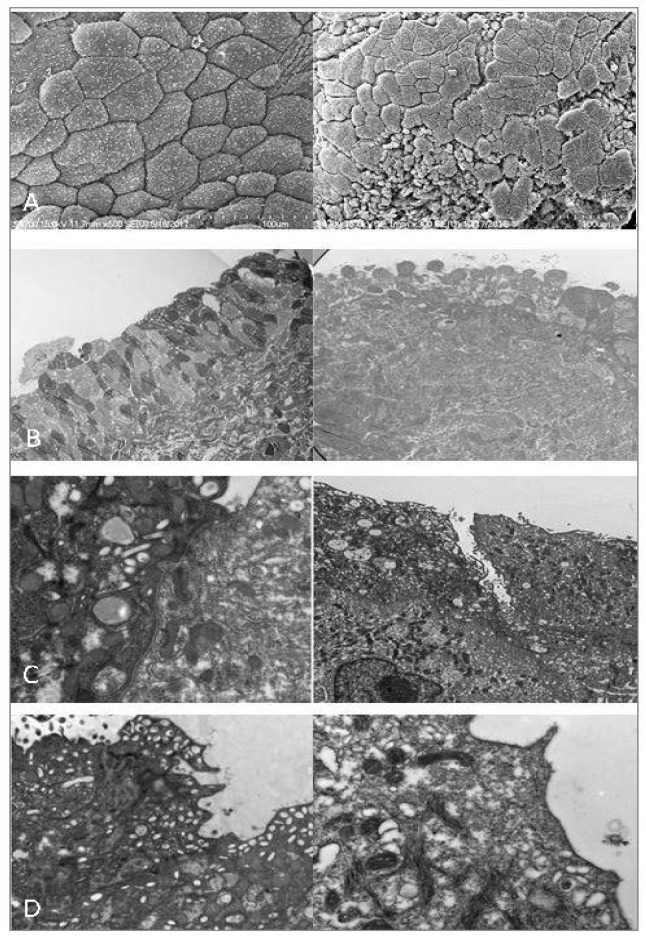
The electron microscopic findings in (**left**) normal bladder urothelium and (**right**) the bladder from a patient with interstitial cystitis/bladder pain syndrome (IC/BPS). (**A**) Scanning electron microscopy images of normal umbrella cells vs. pleomorphism in IC/BPS. (**B**) Transmission electron microscopy images of normal urothelium vs. defects in urothelial cell layers in IC/BPS. (**C**) Normal umbrella cell tight junction vs. defective tight junction in IC/BPS. (**D**) Normal surface microvilli and vesicles vs. the loss of microvilli and vesicles in an IC/BPS bladder.

**Figure 3 biomedicines-10-02380-f003:**
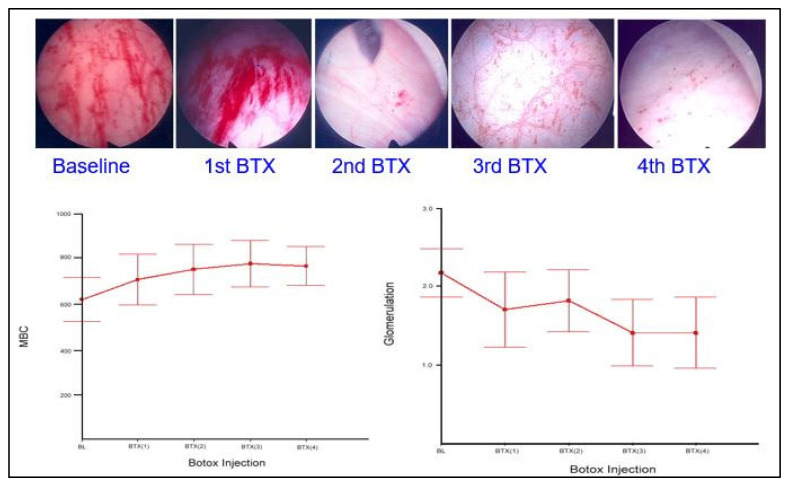
Changes in glomerulations and maximal bladder capacity (MBC) after repeated intravesical Botox (BTX) injections in patients with interstitial cystitis/bladder pain syndrome.

**Figure 4 biomedicines-10-02380-f004:**
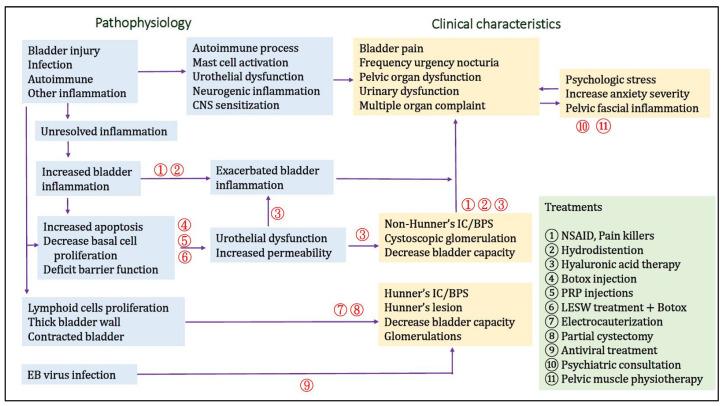
Schematic flow chart of the pathophysiology of interstitial cystitis/bladder pain syndrome (IC/BPS) and the treatments targeting at individual pathophysiology. The number of treatments showing in the diagram indicates the site of treatment targeting at the possible pathophysiology of IC/BPS.

## Data Availability

Not applicable.

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
