# Peer review of "Current Understanding of the Pathophysiology and Novel Treatments of Interstitial Cystitis/Bladder Pain Syndrome"

_biomedicines, 2022, doi:10.3390/biomedicines10102380_

Round 1

Reviewer 1 Report

In this review article, the authors described the current understanding of interstitial cystitis/bladder pain syndrome (IC/BPS) with regard to the pathophysiology, the treatment of HIC and NHIC, and the results of clinical RCT trials. Although the authors try to summarize the present status of knowledges on pathogenesis and standard therapies based on the pathology typing, there are fundamental defects in the manuscript as listed below.

Major points:

1.    The organization and arrangement of the sections should be reconsidered. The authors started the review from the pathophysiology of IC/BPS, and then moved to the treatment of the disease, however, the sections were mixed in the order.  In some parts, the description on pathophysiology and the clinical treatment was mixed up mutually even though the treatments were dependent on the features of pathology.

2.    The sentence describing the fundamental pathophysiology of IC/BPS was repeated several times and redundant, i.e., urothelium injury, suburothelial inflammation, inflammatory cell infiltration (germinal center formation), chronic inflammation, and chronic scar formation in the suburothelium.  

3.    To facilitate the reader’s understanding, I strongly recommend the authors to draw the schematic pictures of pathophysiology of IC/BPS and their flow chart, distinguishing the histological, neural and CNS-related events.

4.    The numbering of the sections was aberrant.

5.    Recent animal model experiments of IC suggest the involvement of HMGB1, an important DAMP, and other molecules in the pathogenesis and pain response. The authors should refer to the literatures.   

Minor points:

1.    Figure 4 (line 376) is missing.

2.    Page 10, lines 413-420: should be arranged.

3.    Reference. 28: The name of first author should be corrected.

Author Response

In this review article, the authors described the current understanding of interstitial cystitis/bladder pain syndrome (IC/BPS) with regard to the pathophysiology, the treatment of HIC and NHIC, and the results of clinical RCT trials. Although the authors try to summarize the present status of knowledges on pathogenesis and standard therapies based on the pathology typing, there are fundamental defects in the manuscript as listed below.

Major points:

  1. The organization and arrangement of the sections should be reconsidered. The authors started the review from the pathophysiology of IC/BPS, and then moved to the treatment of the disease, however, the sections were mixed in the order.  In some parts, the description on pathophysiology and the clinical treatment was mixed up mutually even though the treatments were dependent on the features of pathology.

Reply: Thank you for the comments. The manuscript has been revised to an order following the sequence of clinical disease, pathophysiology, novel treatments. The duplicating paragraphs regarding the pathophysiology of IC/BPS have been deleted. We attempted to combine pathophysiology and treatment targeting that pathophysiology in each section, so that the readers could understand the aim of treatment.

  1. The sentence describing the fundamental pathophysiology of IC/BPS was repeated several times and redundant, i.e., urothelium injury, suburothelial inflammation, inflammatory cell infiltration (germinal center formation), chronic inflammation, and chronic scar formation in the suburothelium.

Reply: Thank you for the comments. The manuscript has been revised to an order following the sequence of clinical disease, pathophysiology, novel treatments. The duplicating paragraphs of the pathophysiology of IC/BPS have been deleted.

  1. To facilitate the reader’s understanding, I strongly recommend the authors to draw the schematic pictures of pathophysiology of IC/BPS and their flow chart, distinguishing the histological, neural and CNS-related events.

Reply: Thank you for the comments. A schematic flow chart of the pathophysiology, clinical characteristics, and potential treatment targeting at each pathophysiology has been added (Figure 4) to facilitate readers’ understanding.

  1. The numbering of the sections was aberrant.

Reply: The numbering of the sections have been revised and rearranged, accordingly.

  1. Recent animal model experiments of IC suggest the involvement of HMGB1, an important DAMP, and other molecules in the pathogenesis and pain response. The authors should refer to the literatures.

Reply: Thank you for the comment. We have added some statements related to the mechanism of bladder pain in IC/BPS in the text. (Lines 265-273)

Minor points:

  1. Figure 4 (line 376) is missing.

Reply: Figure 4 listed at original line 376 has been deleted.

  1. Page 10, lines 413-420: should be arranged.

Reply: Thank you, Text in lines 413-421 has been rearranged (Lines 417-425).

  1. 28: The name of first author should be corrected.

Reply: Thank you. The name of first author in reference 28 has been corrected as Clemens JQ.

Reviewer 2 Report

The authors describe in a comprehensive way all current therapeutic modalities for the treatment of interstitial cystitis/bladder pain syndrome. The introduction is correct, and the article is well conducted. The subtitle “What is interstitial cystitis?” is unnecessary for the knowledgeable reader as it is expected to know the subject of his interest. Some statements are redundant (“Because the underlying pathophysiology is unclear,” line 31 and “The pathophysiology of IC/BPS is still undetermined; line 320, etc. ). One such statement in line 31 is sufficient. The authors should find such redundant places and avoid redundancies in the article.

All therapies are explained in a concise manner and backed with relevant references. Figures a relevant for the illustrations of the article t. Conclusions are relevant to the subject

Author Response

The authors describe in a comprehensive way all current therapeutic modalities for the treatment of interstitial cystitis/bladder pain syndrome. The introduction is correct, and the article is well conducted. The subtitle “What is interstitial cystitis?” is unnecessary for the knowledgeable reader as it is expected to know the subject of his interest. Some statements are redundant (“Because the underlying pathophysiology is unclear,” line 31 and “The pathophysiology of IC/BPS is still undetermined; line 320, etc. ). One such statement in line 31 is sufficient. The authors should find such redundant places and avoid redundancies in the article.

All therapies are explained in a concise manner and backed with relevant references. Figures a relevant for the illustrations of the article t. Conclusions are relevant to the subject

Reply: Thank you for the comments. We have deleted the subtitle “1. What is interstitial cystitis?”. The redundant statements of the uncertain pathophysiology stated in original Lines 320 have been deleted and revised. Because this review article aims to report the current understanding of IC/BPS and novel treatment targeting at the pathophysiology of IC/BPS, some statements are still necessary ahead of the section of treatment paragraphs. We have rearranged the text and make them easily to understand. We have also tried our best to shorten the redundant descriptions.

Round 2

Reviewer 1 Report

The authors reorganized the review according to my suggestions. I think that the manuscript has been improved by introducing the conceptual flow chart in the review. 

Number 5. Conclusion should be 9. Conclusion.